# Bringing SEM and MSI Closer Than Ever Before: Visualizing *Aspergillus* and *Pseudomonas* Infection in the Rat Lungs

**DOI:** 10.3390/jof6040257

**Published:** 2020-10-30

**Authors:** Tereza Juříková, Dominika Luptáková, Olga Kofroňová, Anton Škríba, Jiří Novák, Helena Marešová, Andrea Palyzová, Miloš Petřík, Vladimír Havlíček, Oldřich Benada

**Affiliations:** 1Institute of Microbiology of the Czech Academy of Sciences, Vídeňská 1083, 142 20 Prague, Czech Republic; tereza.jurikova@biomed.cas.cz (T.J.); dominika.luptakova@biomed.cas.cz (D.L.); kofra@biomed.cas.cz (O.K.); anton.skriba@biomed.cas.cz (A.Š.); jiri.novak@biomed.cas.cz (J.N.); maresova@biomed.cas.cz (H.M.); palyzova@biomed.cas.cz (A.P.); 2Department of Genetics and Microbiology, Faculty of Science, Charles University, Viničná 5, 128 44 Prague, Czech Republic; 3Institute of Molecular and Translational Medicine, Faculty of Medicine and Dentistry, Palacký University, Hněvotínská 5, 779 00 Olomouc, Czech Republic; milos.petrik@upol.cz; 4Department of Analytical Chemistry, Faculty of Science, Palacký University, 17. listopadu 12, 771 46 Olomouc, Czech Republic; 5Faculty of Science, J. E. Purkyně University, České mládeže 8, 400 96 Ústí nad Labem, Czech Republic

**Keywords:** fixation, scanning electron microscopy, matrix-assisted laser desorption/ionization mass spectrometry imaging, bacteria, fungi, rat lung tissue

## Abstract

A procedure for processing frozen rat lung tissue sections for scanning electron microscopy (SEM) from deeply frozen samples initially collected and stored for matrix-assisted laser desorption/ionization mass spectrometry imaging (MALDI-MSI) was developed. The procedure employed slow thawing of the frozen sections while floating on the surface and melting in a fixative solution. After the float-washing step, the sections were dehydrated in a graded ethanol series and dried in a critical point dryer. The SEM generated images with well-preserved structures, allowing for monitoring of bacterial cells and fungal hyphae in the infected tissue. Importantly, the consecutive nonfixed frozen sections were fully compatible with MALDI-MSI, providing molecular biomarker maps of *Pseudomonas aeruginosa*. The protocol enables bimodal image fusion in the in-house software CycloBranch, as demonstrated by SEM and MALDI-MSI.

## 1. Introduction

Imaging of human and animal tissues, individual cells, and subcellular structures plays a key role in biomedicine and cell biology research. A combination of data recorded by various imaging microscopy platforms has been developed into correlative approaches, such as correlative light electron microscopy [1], or into the correlation of cryogenic light microscopy and cryofocused ion beam scanning electron microscopy of frozen vitreous cells [2].

Merging visual morphology information with molecular or elemental mass spectrometry data is another challenge. Mutually different scales in lateral resolution and sampling with compatible sample preparation represent the main hurdles for scanning electron microscopy (SEM)/mass spectrometry imaging (MSI) data fusion. From a mass spectrometry point of view, the most common MALDI-MSI method allows for label-free mapping of molecules, including proteins, peptides, lipids, metabolites, and small molecules, directly in tissue. The physicochemical information in a single spot may involve the molecular formula, relative ion abundance, and organic structure-specific product ion mass spectrum. To reveal structure- and function-specific information, it is possible to combine MALDI-MSI molecular maps of tissue sections with classical histology [3,4] and fluorescence microscopy [5,6]. The lateral resolution of MALDI-MSI molecular maps ranges from tens of micrometers to five micrometers [7], and unique setups can reach 1 μm [8]. The actual lateral resolution limit stems from the number of neutrals being desorbed during the desorption event combined with the compound ionization efficiency. Most often, elements or abundant lipids can be visualized in accordance with the selected ionization technique [3,4].

Analogous to MSI, wide-field optical microscopy also does not deliver images with high lateral resolution. On the other hand, SEM is capable of visualizing large samples with unprecedented resolution in a complex environment and a considerable depth of field [9] to show bacterial biofilms or bacterial cell surface details such as flagella and pili [10]. However, the preparation of biological specimens for SEM may be complicated [11], and sample chemical fixation is needed for stabilization and structural preservation. Commonly used glutaraldehyde or paraformaldehyde [12] can interfere with MALDI-MSI analysis [13] or with protein mass spectrometry in general [14,15]. Therefore, protocols with vacuum-dried [16,17,18] or lyophilized [19] frozen tissue sections were developed.

There is no information in the literature on SEM of the same or consecutive tissue sections combined with MALDI-MSI. SEM analysis has been used to monitor the quality of the matrix deposition procedure [20,21] and its influence on the laser ablation process [22]. Fincher et al. [23] used SEM as a method complementary to MALDI-MSI in their analytical study of lipids in human skin disease. They confirmed the presence of bacteria along the hair shafts on the sick surface. To bring MSI and SEM closer than ever before, we report a procedure for the processing of frozen sections for SEM from lung specimens initially collected and stored in a deep-frozen state for MALDI-MSI analysis. The optimized protocol produces samples with sufficient structural feature preservation of the lung tissue that was documented on frozen rat lung tissue infected with the bacterium *Pseudomonas aeruginosa* and the fungus *Aspergillus fumigatus*.

## 2. Materials and Methods

### 2.1. Animal Experiments

All animal experiments were conducted following the regulations and guidelines of the Czech Animal Protection Act (No. 246/1992) and with the approval of the Czech Ministry of Education, Youth, and Sports of the Czech Republic (MSMT-21275/2016-2 and MSMT-9487/2019-5) and the institutional Animal Welfare Committees (Faculty of Medicine and Dentistry of Palacký University in Olomouc). Animal studies were performed using female 2–3-month-old Lewis rats (Envigo, Horst, The Netherlands). During infection models, research staff followed the general guidelines for the protection (European Parliament and the Council, 2000/54/EC, [24]). This included the strict use of protective mask FFP2/KN95, which is N95 HEPA respirator equivalent.

Fungal infection was studied in a rat model, as described previously [25]. Briefly, neutropenia was induced by intraperitoneal injections of 75 mg/kg of the cyclophosphamide (DNA-alkylating agent, Endoxan, Baxter, Prague, Czech Republic) five and one days before inoculation. *A. fumigatus* strain CCF 1059 infection was established by the intratracheal administration of *Aspergillus* spore suspension (100 μL) at a concentration of 1 × 10^8^ spores/mL. Alternatively, the rats were infected intratracheally with *P. aeruginosa* strain ATCC 15692 100 μL of 1 × 10^8^ CFU/mL under 2% isoflurane anesthesia (Forane, Abbott Laboratories, Abbott Park, IL, USA) to minimize animal suffering according to the protocol [26]. The animals were euthanized by exsanguination, and the lungs were immediately dissected out and deeply frozen or aldehyde fixed.

### 2.2. Basic Histology of Infected Lung Tissues

Dried, thaw-mounted, 15 μm-thick frozen sections (Leica Microsystems GmbH, Wetzlar, Germany) were used in two histological staining protocols. *Aspergillus* hyphae in sections were stained by Grocott’s methenamine silver (GMS) [27] using an HT100A silver stain modified kit (Sigma-Aldrich, Prague, Czech Republic). Eosin Y counterstained (VWR Chemicals, Stříbrná Skalice, Czech Republic) and dehydrated sections (ethanol, 95, and 100%) were cleared in xylene and mounted in DPX medium (Sigma-Aldrich, Prague, Czech Republic). Alternatively, modified [28] Gram staining was used for specific *P. aeruginosa* detection. All reagent solutions were prepared from analytical grade chemicals (LaChema, Brno, Czech Republic) and double-distilled water. Safranin (Geigy, Basel, Switzerland) counterstained dehydrated sections (ethanol, 95, and 100%) were cleared in xylene and mounted in DPX medium. Permanent histological mounts of the whole-lung lobe slices were scanned in Minolta DiMAGE Scan Multi PRO (Konica Minolta, Inc., Tokyo, Japan) at 4800 dpi.

### 2.3. Optical Microscopy

GMS-stained sections were examined under a DN45 light microscope (LAMBDA PRAHA Ltd., Prague, Czech Republic) equipped with a Canon EOS 700D Digital SLR camera (Canon, Inc., Tokyo, Japan). The camera’s built-in white balance correction was used for all recorded images. Spatial calibration of the recorded images was performed with the Fiji software suite [29] using a stage (objective) micrometer. Gram-stained lung sections were examined in a Leica DM6000 light microscope equipped with a Leica DFC490 color CCD camera (Leica Microsystems, Wetzlar, Germany). Original images in Leica image file format (.lif) were exported in TIFF format using proprietary Leica software (LAS X Life Science, v. 3.7.1.21655) or Fiji plugins. The associated image metadata were retrieved by OME Bio-Formats [30] Fiji plugins.

### 2.4. Float-Fixation Procedure for Whole Lung Lobe Frozen Sections

A six-well plate (Nunc Non-Treated Multidishes, Thermo Fisher Scientific, Brno, Czech Republic) was filled with 2 mL of fixative solution (3% glutaraldehyde, 0.1 M cacodylate buffer, pH 7.4) and frozen in the freezer at −20
${}^{\circ }C$. Before cryosectioning, the six-well plate and deep-frozen lung lobes were conditioned in a cryo-chamber at −15
${}^{\circ }C$. Then, 25 μm–100 μm-thick frozen sections of the lung tissue were carefully placed onto the solid surface of the fixative solution with one section per well (Figure 1, green path). The lid-covered plate was transferred into the refrigerator and maintained at 4 ${}^{\circ }C$ until all the fixative solution completely melted, usually overnight. During the thawing process, the sections continued to float on the fixative solution and were gently fixed by glutaraldehyde diffusion. Subsequently, the sections were transferred onto the cooled cacodylate buffer level and allowed to float for 20 min three times. Washed sections were mounted onto wet coverslips and wrapped in fine copper wire mesh for easier handling. Then, the whole mounts were dehydrated in a graded ethanol series (25, 50, 75, 90, 96, 100, 100%, 20 min each) and dried in a critical point dryer. Dried sections were mounted onto aluminum specimen stubs using ultrasmooth carbon discs (SPI Supplies, Structure Probe, Inc., West Chester, PA, USA).

### 2.5. Scanning Electron Microscopy

All tissue samples were sputter-coated with 3 nm of platinum in high-resolution Turbo-Pumped Sputter Coater Q150T (Quorum Technologies Ltd, Ringmer, UK) before their examination under FEI Nova NanoSEM 450 field emission gun scanning electron microscope (Thermo Fisher Scientific, Brno, Czech Republic). The Navigation Montage option of the SEM software, v. 6.3.4.3233 (Helios NanoLab, Thermo Fisher Scientific, Brno, Czech Republic) was used to map whole tissue sections at low resolution. Final SEM analyses were performed at acceleration voltage ranging from 2 kV to 5 kV and spot size 3 using Everhart–Thornley Detector (ETD), Circular Backscatter Detector (CBS), and Through the Lens Detector (TLD). If specimen charging was experienced, a beam deceleration mode combined with the magnetic immersion final lens [31,32,33] was used.

### 2.6. Image Data Processing

Proprietary software delivered with microscopes was used unless otherwise stated. In some cases, Fiji software suite, a clone of ImageJ, which is an open source image processing program [35,36], was also utilized. No image processing was applied to the images from SEM with the exception of SEM navigations images and stereo pairs for R-GB anaglyphs.

### 2.7. MALDI-MSI and SEM on Consecutive Sections

Liquid nitrogen snap-frozen rat lung tissue infected with *P. aeruginosa* was cryosectioned, thaw-mounted onto indium-tin-oxide (ITO) glass slides, and vacuum-dried in a desiccator (Figure 1, black path). The slide was then covered with α-cyano-4-hydroxycinnamic acid (CHCA) MALDI matrix (Thermo Fisher Scientific, Czech Republic) in the SunCollect spraying device (SunChrom, Friedrichsdorf, Germany). Before MALDI sampling, the slide was scanned in an optical scanner. Mass spectra were acquired on a SolariX 12T Fourier Transform Ion Cyclotron Resonance (FT-ICR) spectrometer (Bruker Daltonik GmbH, Bremen, Germany) equipped with a 2 kHz laser. Compound dereplication and image fusions were performed by CycloBranch software, v. 2.0.8 (Available online: https://ms.biomed.cas.cz/cyclobranch/ (accessed on 8 May 2012)) [37].

## 3. Results

### 3.1. Optical Microscopy of Lung Tissue with a Fungal or Bacterial Infection

By using GMS staining, the eccentric growth of fungal hyphae in the alveolus is shown in the dark in Figure 2A. The faint pink damaged structure of the alveolar space was counterstained by eosin Y, making bright red blood cell clusters visible. Typical stages of the *A. fumigatus* infectious life cycle, e.g., dissemination, uncontrolled hyphal growth through the pulmonary parenchyma, and severe tissue damage [38], were also detected in the infected rat lung tissue. Dichotomous branching of the hyphae at acute angles and individual septa, typical for *Aspergillus* in the infected tissues [39], were visible at higher magnification (Figure 2C).

Although it was possible to detect individual *P. aeruginosa* cells in the Gram-stained lung sections, their morphology could barely be judged (Figure 2B). As the bacterial cells were hardly detectable at medium magnification, the rod-shaped bacteria inside the alveolus were recorded using an oil immersion objective (Figure 2D). The images illustrate the standard data achieved in a conventional laboratory with typical staining procedures, here representing the current limits of routine optical microscopy.

### 3.2. Standard Tissue Thaw-Mounting in MSI Hampers Potential SEM Analysis

Vacuum-dried sections from deep frozen lung lobe mounted onto glass slide were used for SEM imaging (Figure 1, red path). As the sections were mounted onto non-conductive glass support, we experienced sample charging when the samples were imaged in secondary electrons. Therefore, backscattered electron mode was instead used for image acquisition.

It was believed that thaw-mounting keeps the sample intact from a (bio)chemical point of view [17,18], which could be true, namely, for lipid features. We showed that, after the thaw-mounting step, the surface of the vacuum-dried sections was covered with a distinct layer of dried colloids of intra- and extracellular origin, which did not interfere with the MALDI-MSI analysis but made SEM analysis useless. The SEM results were typically worse than those from an optical microscope, regardless of whether the sample was coated with the matrix.

### 3.3. SEM Float Fixation/Washing Procedure Preserved the Tissue Structure

Surface tension forces allowing the frozen section to float on the fixation and washing solutions were the critical factors of our procedure. In this new procedure, glutaraldehyde fixed the sections during thawing on the surface of the melting frozen fixative solution at 4 ${}^{\circ }C$. The remaining glutaraldehyde was then washed away by floating the sections on the surface of a cooled cacodylate buffer before further processing for SEM (Figure 1, green path).

### 3.4. SEM Imaging of Infected Tissues and Data Fusion with MALDI-MSI

All infected tissues were processed by the float-fixation/washing method. Figure 3 shows the lung lobe section images at low magnification with branched or cross-sectioned fungal hyphae (Figure 3A,C, respectively). The tip detail or a cross-section of fungal hyphae at higher magnification is illustrated in Figure 3B,D, respectively. The diameters of the imaged *Aspergillus* hyphae ranged from 1.8 μm to 3.8 μm with an average value of 2.7
μm, which agreed well with literature data [40]. GMS stained a consecutive section was typically employed to find fungal hyphae’s specific location in the prepared SEM specimen (Appendix A). It was practically impossible to efficiently discover the *Aspergillus* hyphae in such a large tissue section without it.

Figure 3E–H is dedicated to bacterial data indicating the cells of *P. aeruginosa* in the alveolar space. The arrows point to a cluster of bacterial cells, apparently formed by active bacterial growth in the tissue (Figure 3G). The damaged tissue and elastin and collagen fibers along with the bacteria are illustrated in Figure 3H. The individual panels precise locations in the specific area of the whole section are presented in the Appendix A. It confirmed the *P. aeruginosa* cells in the lobe’s distinct part indicated by the pyoverdine E protonated molecule map (Figure 4B,C).

As the flow fixation/washing procedure was capable of preserving fungal hyphae and bacterial cells in the frozen sections of infected tissue, the SEM images were merged with MALDI-MSI data from a consecutive section. Bacteria were detected on the alveolar surface in the section part to which pyoverdine E [41] was mapped by MALDI-MSI (Figure 4). Initially, a scanned image of the slide served for the orientation of the MALDI-MSI results (Figure 4A). Then, the pyoverdine E protonated molecule map, i.e., [C${}_{55}$H${}_{84}$N${}_{18}$O${}_{21}$+H]${}^{+}$, from the mass spectrometry imaging imzML data format was merged with a scanned slide image in CycloBranch software [37] and manually matched with SEM navigation image (Figure 4B,C). Next, all SEM images of the set recorded from the region of the section with a reliable pyoverdine signal were automatically aligned with the navigation image in CycloBranch software using the TIFF file metadata. The two squares indicated in Figure 4D illustrate the 50 μm × 50 μm area irradiated by the laser beam containing the rod-shaped bacterial cells.

## 4. Discussion

Round-shaped irradiated areas (a Gaussian beam profile [22]) with a diameter of ten micrometers are usually collected for bimodal fusion of MALDI-MSI with optical microscopy [37]. At this lateral resolution, the highly abundant siderophore molecules that are microbe-characteristic can be detected. One of the very first reports was ferricrocin molecular imaging in rat lungs heavily infected with *A. fumigatus* [25]. However, extensive hemorrhage and considerable tissue damage caused by the lasers destroyed most of the morphological details [42]. One must consider the dynamic range of the MALDI-MSI experiment. In theory, it is possible to scale down the MSI experiment to a single cell or even to the subcellular scale [43,44]. However, only the most abundant, i.e., functionally or diagnostically fewer interesting molecules, can then be visualized.

Optical microscopy has not been able to provide sufficiently detailed structural features but served as a quick tool for examination of the extent of fungal or bacterial infection in rat lung tissue. Nevertheless, eccentric growth of fungus in alveolus (Figure 2A) and other characteristic morphological features for *Aspergillus* in the infected tissues [39,40,45] are clearly documented at higher magnification (Figure 2C). The quality of the SEM images recorded from samples prepared by float-fixation method corresponded to the literature data [46]. Importantly, for the infected tissue studies, the sections handled by the float/washing protocol did not exhibit excessive flattening (Appendix A, with 3D R-GB anaglyphs), as was observed for the vacuum-dried samples.

At a higher magnification than that shown in Figure 4D, a distinct displacement of the SEM image can be observed. The bacterial density does not precisely reflect the molecular intensity of the pyoverdine signal, which is caused by different processing of the frozen sections for both imaging methods (Figure 1). A slight alteration of the tissue structure in some parts of 20 μm to 40 μm-thick sections can be observed more frequently than that in thicker cuts (50 μm to 100 μm), which better withstand physical stress during processing.

To conclude, preservation of the lung tissue structure in the samples infected with *A. fumigatus* or *P. aeruginosa* was sufficient for detailed SEM analysis of alveolar space colonized with both microorganisms. The uncomplicated flow-fixation/washing method is not limited to lung tissue and can be used to process frozen sections from other soft biological tissues. It does not require any specialized tools or sophisticated equipment and is feasible for any standard laboratory with occasional access to an electron microscopy facility. Of final note, the float-fixation procedure worked even for relatively large sections with an area exceeding 50 mm^2^. In other words, sections through the whole rat lung lobe can be processed.

## Figures and Tables

**Figure 1 jof-06-00257-f001:**
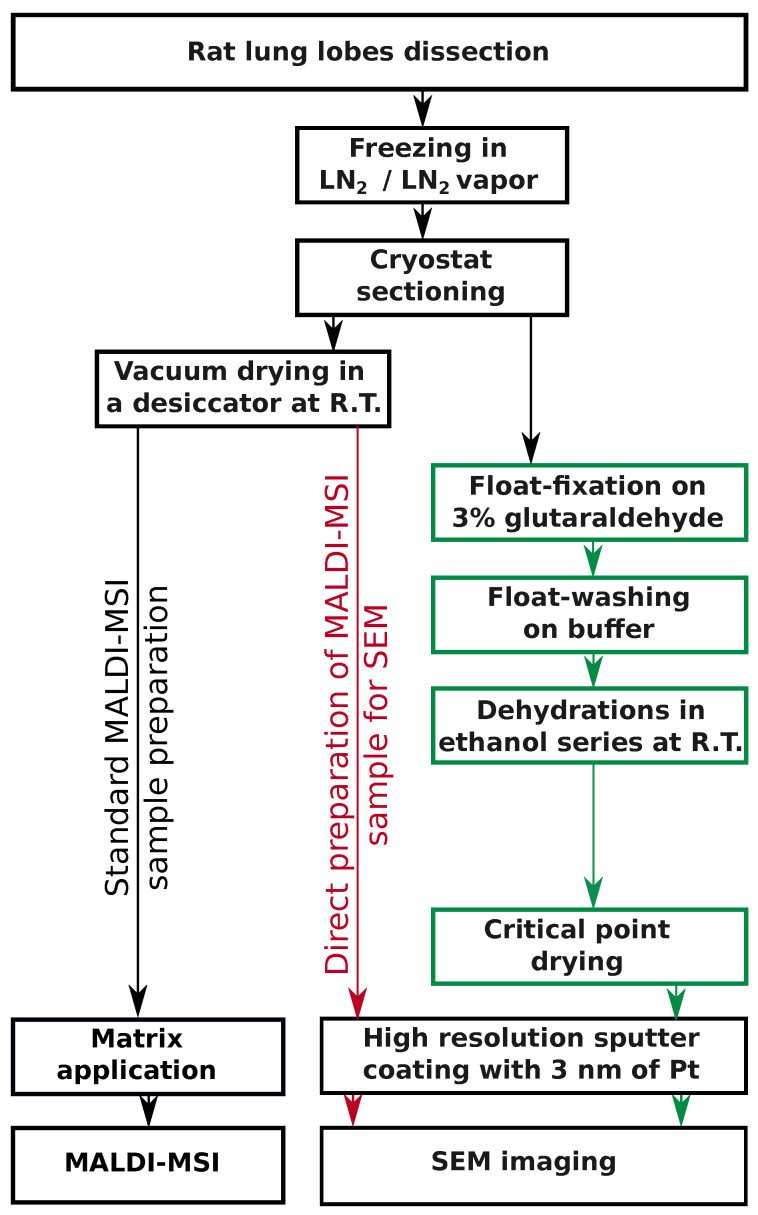
An overview of lung sample processing for SEM and MSI. The individual steps of the float-fixation/washing procedure are shown in green. MALDI-MSI samples directly prepared for SEM are enhanced in red. Additionally, parallel processing of frozen vacuum-dried lung tissue sections for MALDI-MSI analysis is shown in black in the left part of the panel. Color coding was selected according to AAAS color palette [34]. LN${}_{2}$, liquid nitrogen; R.T., room temperature.

**Figure 2 jof-06-00257-f002:**
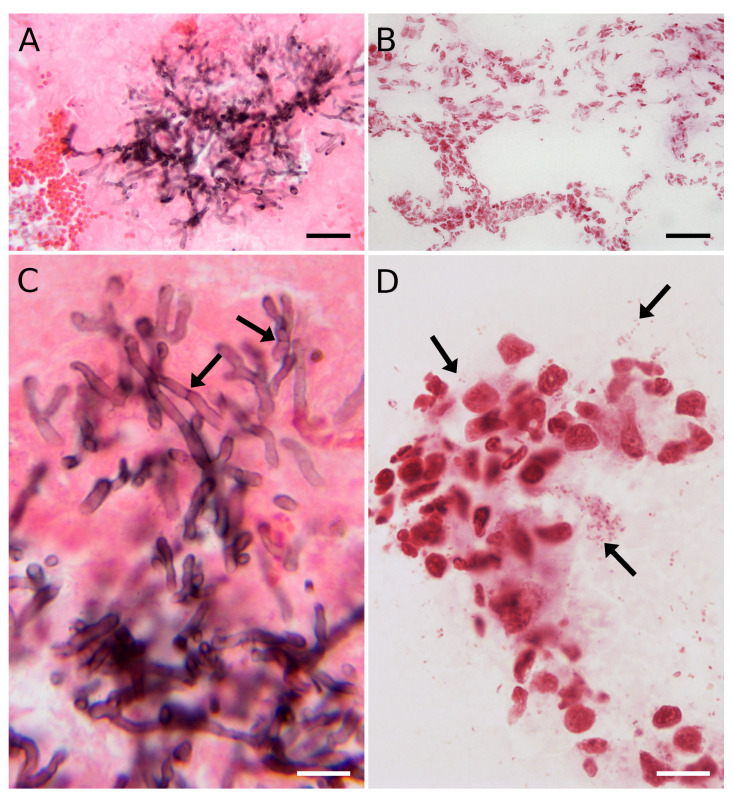
Histological examination of GMS-stained rat lungs infected with *A. fumigatus* hyphae (**A**,**C**) and Gram-stained *P. aeruginosa* cells (**B**,**D**). (**A**) Spreading of the fungus into the alveolar space. (**B**) Bacterial infection, 40× objective. (**C**) Septate (arrows) and branching hyphae. (**D**) Individual bacterial cells (arrows), single or in distinct clusters imaged with a 100× immersion objective. (**A**) A Leica built-in white balance compensation and (**D**) a custom white balance compensation were used. (**B**,**C**) The camera’s built-in white balance correction was used. Scale bars: 30 μm (**A**,**B**) and 10 μm (**C**,**D**).

**Figure 3 jof-06-00257-f003:**
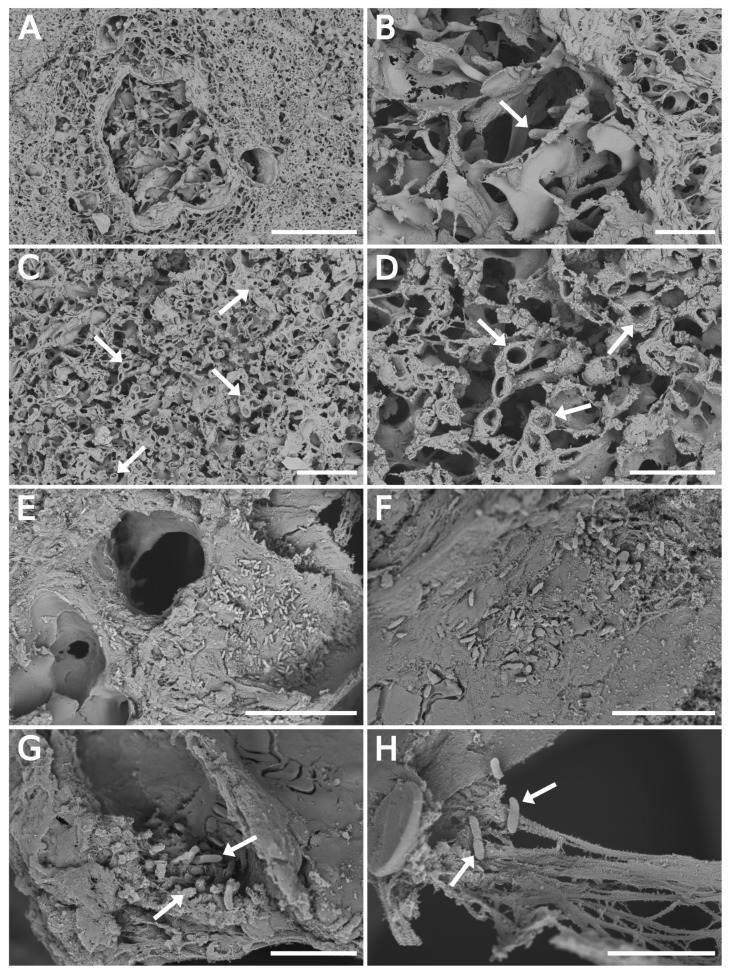
SEM backscattered electron imaging of float-fixed cryosections from rat lung lobes. (**A**–**D**) Lung tissue infected with *A. fumigatus*, 100 μm-thick cryosections. (**A**,**B**) Fungal hyphae invading the pulmonary parenchyma. The arrow points to the hyphae tip. (**C**,**D**) Cross-sectioned hyphae (arrows) in the alveolar space. (**E**–**H**) Lung tissue infected with *P. aeruginosa*, 50 μm-thick cryosections. (**E**,**F**) Overview images of alveolar space with bacteria. (**G**,**H**) Bacteria in damaged tissue. Arrows point to individual bacterial cells. Primary magnification of SEM images were: (**A**) 1000×; (**B**,**C**) 3500×; (**D**) 10,000×; (**E**) 6500×; (**F**) 12,000×; (**G**) 20,000×; and (**H**) 25,000×. Scale bars: 100 μm (**A**); 20 μm (**B**,**C**,**E**); 10 μm (**D**,**F**); and 5 μm (**G**,**H**).

**Figure 4 jof-06-00257-f004:**
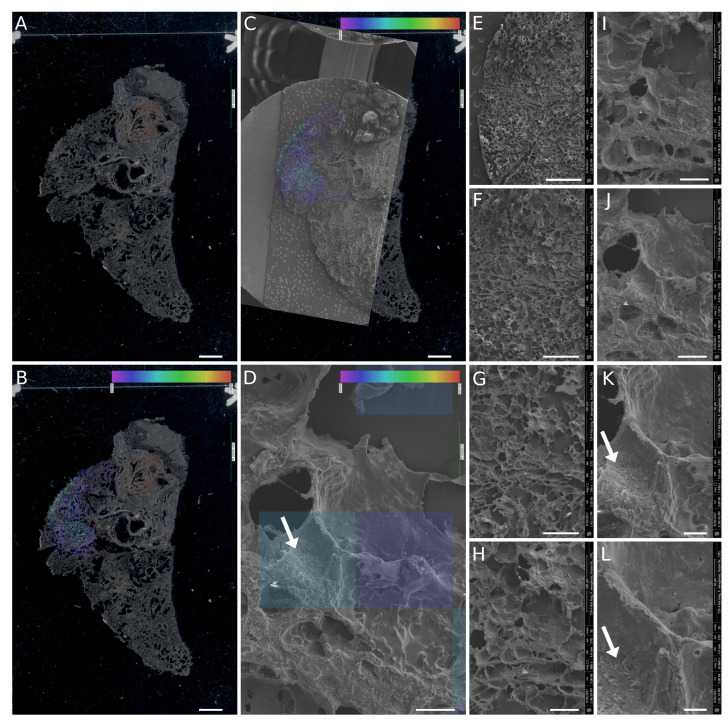
The fusion of MALDI-MSI and SEM imaging of rat lung tissue infected with *P. aeruginosa*: (**A**) optical scan of the ITO glass slide with the vacuum-dried section; (**B**) optical scan image with MALDI-MSI pyoverdine molecular map in the overlay; (**C**,**B**) fusion with the SEM navigation image; and (**D**) zoomed-in part of the fused images showing rod-shaped bacteria on the alveolar surface (white arrow, also demonstrated in (**K**,**L**)). Image fusion was generated from an SEM image (**J**) with a lateral resolution of 100 nm per pixel and a part of the pyoverdine molecular map. The size of the transparent blue squares in the overlay corresponds to the lateral resolution of 50 µm per pixel. (**E**–**L**) A set of the original SEM images used for detailed characterization of the area with a positive pyoverdine signal. Primary magnifications of the SEM images were: (**E**) 85×; (**F**) 170×; (**G**) 340×; (**H**) 680×; (**I**) 1360×; (**J**) 2720×; (**K**) 5440× and (**L**) 10,880×. Scalebars: (**A**–**C**) 2 mm; (**D**) 20 μm; (**E**) 1 mm; (**F**) 500 μm; (**G**) 250 μm; (**H**) 100 μm; (**I**) 50 μm; (**J**) 25 μm; (**K**) 10 μm; and (**L**) 5 μm.

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
