# Peer review of "Bringing SEM and MSI Closer Than Ever Before: Visualizing Aspergillus and Pseudomonas Infection in the Rat Lungs"

_jof, 2020, doi:10.3390/jof6040257_

Round 1

Reviewer 1 Report

The authors present a method paper for SEM imaging and MALDI-MSI for both A. fumigatus and Pseudomonas aeruginosa. Its a mainly technical description for a method that is widely used but first time published for A. fumigatus. The paper is clear and well structured. The methods are adequate. To my understanding it can be published as it stands, as its mainly a method paper.

Author Response

Dear Reviewer,
Thank you for your positive thought about our manuscript. It is actually encouraging to read your words on it.

Oldřich Benada
Vladimír Havlíček

Reviewer 2 Report

This is the manuscript on the protocol that enables bimodal image fusion in the in-house software CycloBranch, as demonstrated by SEM and MALDI-MSI. This is an interesting manuscript.

I wonder if the authors could observe the vacuumed-dried section with SEM without sputter coating. If it is possible, complete CLEM between SEM and MSI may be possible with the same cryostat section. From my experience, you may avoid the charging effect with SEM backscattered electron mode without sputtering. The authors need to show the SEM image of the vacuumed-dried section without sputter coating. 

Author Response

This is the manuscript on the protocol that enables bimodal image fusion in the in-house software CycloBranch,as demonstrated by SEM and MALDI-MSI. This is an interesting manuscript.

Dear Reviewer,Thank you for your positive opinion on our manuscript and many thanks for your valuable comments.

I wonder if the authors could observe the vacuumed-dried section with SEM without sputter coating. If it is possi-ble, complete CLEM between SEM and MSI may be possible with the same cryostat section. From my experience,you may avoid the charging effect with SEM backscattered electron mode without sputtering. The authors need to show the SEM image of the vacuumed-dried section without sputter coating.

We have spent much time with air-dried unfixed lung lobe frozen sections and the possibility of CLEM between SEM and MALDI/MSI. The main problem of the lung lobe frozen section is in the air-drying artifacts. In our hands,all air-dried (vacuum-dried) sections were useless for SEM analysis. The SEM of air-dried (vacuum-dried) sections on glass (or ITO-glass) showed a layer or overlay on top of the dried sections created likely by dried cytosol orextracellular fluids released from the frozen sections during the thaw mounting process. Even specific structural details as alveolar ducts and individual alveoli were buried beneath that thick cover. This cover did not interfere with MSI as MALDI analysis is a destructive method ablating a part of the section by a laser pulse. From our pointof view, this represents a crucial obstacle in real CLEM between SEM-MALDI/MSI. We also tried freeze-drying(lyophilization) of the frozen section, but with mixed results. In this case, the frozen sections did not suffer from drying artifacts so much as air-dried ones. However, during the freeze-drying process, the sections were partially released from supporting glass (ITO-glass). Flattening the dried sections back resulted in their damage.

Air-drying (vacuum-drying) also does not interfere with classical histology staining (e.g., Hematoxylin-eosinstaining or Grocott’s methenamine silver staining) because those procedures comprise the steps in which this artificial coat is washed away. We successfully utilized this fact in our previous paper in which we showed detection of Grocott’s methenamine silver-stained fungal hyphae in the rat lung tissue by SEM, including the EDS proof ofsilver and gold in the fungal hyphae walls and combined it with laser ablation inductively coupled plasma mass spectrometry (ICP MS, Pluhacek et al., 2016, doi:10.1002/pmic.201500487; ref. no. 42 in our manuscript). The float-fixation method was fine-tuned on non-infected mice lung lobes when we performed all controls mentioned in your report. However, we did not include it in our paper since it was done on non-infected mouse tissue.

Another problem is with the infectious character of Aspergillus or Pseudomonas infected lung tissue. With aldehyde fixation, it might be minimized, however without it, air-dried (vacuum-dried) lung tissue remains infec-tious. The float-fixation method might thus overcome the problem with the infectious character of the sample.

We just prepared fresh vacuum-dried sections from deply frozen rat lung lobes to fulfill your demand to show uncoated vacuum dried tissue section.

Unfortunately, the FEG cathode has gone just now in our SEM. The repair will take some time, and we cannot give a real estimate of how it will take.

Reviewer 3 Report

The Authors present a very important basic work to map pulmonary infections.

The Methodology is appropriate and reproducible.

Concerning SEM: in my opinion it would be convenient to express the amplification used, in Figure 3, beyond the microorganisms legnth - a single HE image pointing the pulmonary lobule localization of the collected tissue for SEM would be clinically relevant beyond technically informative.

Results and Discussion deserve to be presented apart, in order to follow the research conclusions and to integrate in the actual knowledge. 

Author Response

The Authors present a very important basic work to map pulmonary infections. The Methodology is appropriate and reproducible.

Dear Reviewer,We would like to thank you for your positive opinion on our manuscript and valuable suggestions to improve it.

Concerning SEM: in my opinion it would be convenient to express the amplification used, in Figure 3, beyond the microorganisms legnth - a single HE image pointing the pulmonary lobule localization of the collected tissue for SEM would be clinically relevant beyond technically informative.

We added the following specification of SEM magnification for all SEM images in Figure 3 to the legends:Primary magnification of SEM images were: A –1000×; B and C -3500×; D -10 000×; E -6500×; F -12 000×; G -20 000×, and H -25 000×.

We also prepared two new composed images for Supplementary Materials. In Figure S2, we used panels A,B, C, and D of Figure 3 of the manuscript. We combined them with new six panels with optical microscopy of consecutive sections stained by GMS (panel H) and H&E (panels I and J), the optical scan of the GMS stained section corresponding to the optical microscopy image in panel H (panel E), camera image of sections on SEM support stub (panel F), and SEM navigation image (panel G). This should allow the readers to locate SEM images used in Figure 3 to the lung lobe section.

The similar approach we used for Figure S3 we composed already used panels E, F, G, and H of Figure 3 of the manuscript, documenting Pseudomonas infection with two new SEM navigation images. This should also allow the readers to locate the SEM images of Pseudomonas infection used in Figure 3 correctly to the lung lobe section.

Results and Discussion deserve to be presented apart, in order to follow the research conclusions and to inte-grate in the actual knowledge.

In the manuscript, we split the “Results and Discussion” into two separate parts. We added appropriate text referring to the new figures in the ”Supplementary Materials” to the manuscript’s Results part and also three more references to the newly rewritten Discussion.

Round 2

Reviewer 2 Report

Authors answered my question honestly and sincerely. Manuscript can be accepted in present form.

Reviewer 3 Report

The Authors may congratulate: methodology standing point has been achieved, as well reproducible results to be followed.